# Evaluation of Marine Fisheries Vulnerability in China and Its Spatial Effects: Evidence from Coastal Regions

**Yutong Li [1] and Jianyue Ji [1,2,*]**

1    School of Economics, Ocean University of China, Qingdao 266100, China; liyutong3200@stu.ouc.edu.cn
2    Institute of Marine Development, Ocean University of China, Qingdao 266100, China
*    Correspondence: jijianyue@ouc.edu.cn; Tel.: +86-139-697-95790

**Abstract:** Based on data from 10 coastal regions in China, this study establishes an evaluation index system for marine fisheries using the theoretical framework of the DPSR model. The entropy method is used to calculate the degree of marine fisheries vulnerability in each region of China from 2009 to 2018. The spatial Durbin model (SDM) is also used to analyze the influencing factors and spatial spillover effects of marine fisheries vulnerability from four perspectives of economic efficiency, industrial structure, environmental regulation, and ecological pollution. The results show significant positive direct effects between the economic efficiency, ecological pollution, and vulnerability of marine fisheries. At the same time, there are significant negative effects between the industrial structure, environmental regulation, and vulnerability of marine fisheries. In terms of spatial spillover effects, economic efficiency, environmental regulation, and ecological pollution show positive spatial spillover effects, while the industrial structure shows negative spatial spillover effects. These findings provide a theoretical basis for the sustainable development of marine fisheries in China.

**Keywords:** marine fisheries; vulnerability; DPSR model; SDM; spatial spillover effects

## 1. Introduction

As population grows and living standards improve, so does the demand for food security and abundance. Land agriculture is somewhat saturated due to the depletion of land resources, which results in increased attention being paid to the ocean [1]. Marine fishery products have become important sources of animal protein as well as a healthy food in China. Marine fisheries are a necessary form of human exploitation of marine fishery products and contribute to the dietary structure optimization. China has the world's largest seafood and aquaculture production, and the economic output of this industry continues to grow. However, extensive development has led to increasingly serious problems, such as the excessive exploitation of marine fisheries resources, severe pollution within the ecosystem [2]. Consequently, the vulnerability of marine fisheries has become increasingly evident, posing a severe threat to the sustainable development of China's marine fisheries.

Since Timmerman's introduction of vulnerability in geology in 1981 [3], research on vulnerability has been continuously refined by scholars in various countries and is essential in sustainable development research [4–8]. Most existing research has focused on analyzing and evaluating vulnerabilities in agriculture, water resources, tourism sites, or specific geographical locations. Adu et al. analyzed the vulnerability of farm households to climate change by using the livelihood vulnerability index [9]. Feng et al. analyzed and evaluated the vulnerability of water resources by constructing a water resources vulnerability evaluation index system [10]. Darabi et al. used hierarchical analysis to assess the vulnerability of mountainous landscape ecosystems and proposed strategic planning and solutions for environmental management in the region [11]. Mennella et al. considered a specific domain as a complex system closely related to natural and human resources and assessed

the vulnerability of the specific domain using fuzzy set theory and hierarchical analysis methods [12]. However, as far as the marine fisheries sector is concerned, research on vulnerability is still underdeveloped. In terms of research subjects, vulnerability studies have been conducted for specific types of fisheries, such as small-scale fisheries [13] and squid fisheries [14]; specific seafood fisheries, such as coral communities [15] and tuna and sardines [16]; specific regional fisheries, such as wetlands [17] and coastal regions [18]; and specific perspectives, such as marine catches [19] and fishermen livelihood [20]. In terms of evaluation methods, the most common methods used for marine fisheries vulnerability include the composite index method and productivity susceptibility analysis (PSA). The composite index method is intuitive and flexible. Hughes et al. [21] constructed vulnerability evaluation indicators in the three dimensions of exposure, sensitivity, and adaptive capacity, and developed country-level vulnerability indices to measure the relative vulnerability of coral reef fisheries in each country. Silva et al. [22] constructed a coastal vulnerability index in terms of three components—species vulnerability, ecosystem vulnerability, and adaptive capacity—to assess the ecological vulnerability of coastal fishing communities. The PSA method combines the attributes of species productivity with the attributes of capture susceptibility to quantify a single vulnerability score. Hornborg et al. [23] used the PSA method to assess the potential vulnerability of marine fishing communities in Skagerrak-Kattegat (eastern North Sea). Previero et al. [24] combined the PSA method with scale-intensity consequence analysis (SICA) to evaluate the potential impacts, risks, and vulnerability of coral reef fisheries in biological, environmental, social, and economic terms. In terms of the influencing factors, marine fisheries vulnerability studies have shifted from single to multiple factors. In terms of single factors, Islam et al. [25] developed an analysis of the vulnerability of fisheries in the coastal regions of Bangladesh in the face of climate change. Further, Juan et al. [26] generated vulnerability indices by combining biological characteristics to study the vulnerability of benthic fisheries to trawling activities. In terms of multiple factors, based on the study of the spatial and temporal evolution characteristics of marine fisheries vulnerability, Li et al. [1] analyzed secondary indicators under the two dimensions of sensitivity and coping capacity by using the barrier degree model and classified them into the first, second, and third influencing factors. From a social–ecological complex system perspective, Chen et al. [27] discussed vulnerability under multiple disturbances based on the three dimensions of "exposure–sensitivity–adaptability," arguing that multiple external disturbances from nature and society are still the main drivers of vulnerability formation and that the high intensity of fishing and mariculture are also important reasons for the increase in vulnerability. Chen et al.'s [28] analysis of the correlation between the marine fisheries vulnerability index and the various drivers shows that vulnerability is more highly correlated with internal factors, such as the level of economic development and governance of the system, and less so with external factors.

Based on this, considering the pre-existing studies and the availability of data, 10 coastal regions in China (excluding Shanghai, Hong Kong, Macau, and Taiwan) are selected as the research objects of this paper. These 10 coastal regions can be divided into three regions, namely the Bohai Sea region, the Yangtze River Delta region, and the Pan-Pearl River Delta region. The Bohai Sea region is a vast area surrounded by the entire Bohai Sea and part of the Yellow Sea coastal areas, which means in this study, Tianjin, Hebei, Liaoning, and Shandong are designated to the Bohai Sea region. As an inner sea embraced by continent, Bohai Sea region, with high fertility rate and water quality, was conducive to fisheries management and production. However, in recent years, the development of marine fisheries in the Bohai Rim region has been seriously threatened by the sloppy development of the region and the irrational exploitation of fishery resources, resulting in the continuous deterioration of the ecological environment and other problems. The Yangtze River Delta region, located in the downstream area of the Yangtze River in China, is bordered by the Yellow Sea and the East China Sea. In this study, Jiangsu and Zhejiang belong to the Yangtze River Delta region. Because of their proximity to shallow waters of the continental shelf, they are high-quality habitats for various fishery resources and

production concentration areas for traditional fisheries. However, the value of high-quality fishery resources has gradually declined in recent years, and the proportion of low-value fishery resources has increased. Fujian, Guangdong, Guangxi, and Hainan belong to the Pan-Pearl River Delta region. The region has formed a fishery industry layout with its own characteristics and advantages in the process, and there are obvious complementarities. However, this has also led to the problems of scattered fishery resource layout, depletion of high-quality fish resources, and low technology. In summary, the development of marine fisheries in China's coastal regions is almost always constrained by problems such as resource depletion and environmental degradation [29].

In summary, some scholars have conducted studies on the vulnerability of marine fisheries, laying the foundation for subsequent studies, but there are still some gaps in this literature stream. Most existing studies on marine fisheries vulnerability evaluation explored the degree of fisheries vulnerability from the three dimensions of system, environment, and human coping capacity. Notwithstanding, the correlations among the various influencing factors are not solid, human drivers are excluded, and the deeper causes of marine fishery impacts are not sufficiently reflected. Given this, the Driver–Pressure–State–Response (DPSR) model is introduced to improve the vulnerability evaluation index system for marine fisheries. This model has transparent logical relationships, highlights the stressors and drivers that lead to changes in the state of natural systems, and can reflect the mutual causal relationship between humans and natural systems [30]. Therefore, this study proposes an index evaluation system based on the DPSR model and applies the entropy value method to assess the vulnerability of marine fisheries in the coastal regions of China. Second, due to the mobility of fishery resources and seawater, there is an intertemporal exchange of fishery resources and pollutants between regions. The health status of fishery ecosystems in adjacent regions is also essentially affecting the vulnerability of marine fisheries. However, most existing articles discussed the factors influencing marine fisheries vulnerability in terms of the indicators in the evaluation system, only considering the impact of the influencing factors over time, without considering spatial mobility. To improve the analysis of the factors influencing fishery vulnerability, this study further explores the spatial spillover effects of the vulnerability of marine fisheries based on the evaluation of this vulnerability in each region using Moran's index and the spatial Durbin model (SDM). This study thus provides theoretical support for the sustainable development of marine fisheries in China.

## 2. Materials and Methods

### 2.1. Data Sources

This study considers 10 coastal regions as the research object and selects relevant data for 10 years from 2009 to 2018 as the basis of this study. The data involved in this study are all taken from the China Statistical Yearbook (2009–2018) [31], the China Marine Statistical Yearbook (2009–2018) [32], the China Fisheries Yearbook (2009–2018) [33], the China Environmental Statistical Yearbook (2009–2018) [34], and the China Marine Ecological and Environmental Status Bulletin (2009–2018) [35]. In the evaluation index system, data on GDP per capita, Engel's coefficient, population density, and industrial wastewater discharge are obtained from the China Statistical Yearbook; data on the area affected by storm surge disaster, equivalent pollutant discharges, region of marine protected regions/the total sea region, and investment in marine science and technology are from the China Marine Statistical Yearbook; data on the fishing production/total production, the region of culturable sea, the economic losses from disasters/total economic output of the fisheries, diversity index, number of aquatic fry per capita, species richness index, and number of fisheries law enforcement agencies are from the China Fisheries Yearbook; data on the equivalent pollutant discharges and governance capacity of industrial wastewater treatment facilities are from the China Environmental Statistics Yearbook; data on water quality in offshore waters are from China Marine Ecological and Environmental Status Bulletin. Among the influencing factors, data on economic efficiency and industrial structure are

taken from the China Fisheries Yearbook, data on environmental regulation from the China Statistical Yearbook, and data on ecological pollution from the China Environmental Statistics Yearbook.

*2.2. Analysis Methods*

This study develops an evaluation index system based on the DPSR model. Additionally, the entropy method is used to evaluate the vulnerability of marine fisheries in the coastal regions of China. Entropy is an objective weighting method that determines the degree of dispersion based on the actual value of each indicator. The greater the degree of dispersion, the greater the influence of an indicator on the overall evaluation. The results obtained using the entropy method are more credible and accurate than those obtained by the subjective weighting method. Next, the global Moran's I model tests the spatial autocorrelation of the vulnerability of marine fisheries in each coastal region of China. Finally, the SDM is used to test the spatial and spillover effects of marine fisheries vulnerability in each coastal region of China based on the LM, LR, and Hausman tests.

### 2.2.1. DPSR Model

Chinese scholar Zuo et al. proposed the driver–pressure–state–response (DPSR) model [36]. This study integrates the pressure–state–response (PSR) [37] and driver–pressure–state–impact–response (DPSIR) [38] models and expands the meaning of the original model to emphasize the drivers that enable human activities to affect natural systems. Based on the theoretical foundation of the DPSR model, this study establishes a system of indicators for evaluating the vulnerability of marine fisheries in coastal regions of China. A driver indicates the influence that motivates people to undertake marine fishery activities. Humans are economically and demographically driven to carry out productive activities and draw resources from the fishery ecosystem. Pressure indicates the direct or indirect impact of human activity on marine fisheries. Under pressure, marine fisheries present certain environmental qualities and sustainability in terms of resources. Marine fisheries feed their state back to the human society, which prompts humans to implement specific measures and policies on marine fisheries to improve the existing state. As a result, marine fisheries exhibit a certain degree of vulnerability under the DPSR interactions.

### 2.2.2. Entropy Method

Step 1. Data matrix

With $m$ programs to be assessed and $n$ evaluation indicators, the initial indicator data matrix is formed as $X = (x_{ij})_{m \times n}$:

$X = \begin{bmatrix} x_{11} & \cdots & x_{1m} \\ \vdots & \vdots & \vdots \\ x_{n1} & \cdots & x_{nm} \end{bmatrix}$ where $x_{ij}$ represents the data for the jth indicator of the ith program.

Step 2. Standardize the data

Because each index differs in magnitude, standardization is required to eliminate the influence of the different magnitudes on the results.

For positive indicators

$$\mu_{ij} = \frac{x_{ij} - \min(x_{1j}, x_{2j}, \cdots x_{nj})}{\max(x_{1j}, x_{2j}, \cdots x_{nj}) - \min(x_{1j}, x_{2j}, \cdots x_{nj})} \tag{1}$$

For negative indicators

$$\mu_{ij} = \frac{\max(x_{1j}, x_{2j}, \cdots x_{nj}) - x_{ij}}{\max(x_{1j}, x_{2j}, \cdots x_{nj}) - \min(x_{1j}, x_{2j}, \cdots x_{nj})} \tag{2}$$

where $\mu_{ij}$ is the data for the *j*th indicator of the *i*th program after standardization.

Step 3. Calculate weight $P_{ij}$ of the $i$th program under the $j$th indicator

$$P_{ij} = \frac{\mu_{ij}}{\sum\limits_{i=1}^{m} \mu_{ij}} \tag{3}$$

Step 4. Calculate entropy value $e_j$ of the $j$th indicator

$$e_j = k \sum_{i=1}^{m} P_{ij} \ln(P_{ij}) \tag{4}$$

where k is a constant; $k = -\frac{1}{\ln m}$.

Step 5. Calculate difference coefficient $g_j$

The difference coefficient of the indicator depends on the difference between information entropy $e_j$ of the indicator and 1:

$$g_j = 1 - e_j \tag{5}$$

Step 6. Calculate evaluation index weights

The larger the index weight, the greater the degree of contribution to the evaluation results.

$$\omega_j = \frac{g_j}{\sum\limits_{j=1}^{n} g_j} \tag{6}$$

Step 7. Calculation of the composite score of each program

$$S = \sum_{j=1}^{n} P_{ij} \cdot \omega_j \tag{7}$$

The larger the value of $S$, the better the evaluation result.

### 2.2.3. Spatial Weight Matrix
Proximity Weight Matrix ($W_{0,1}$)

In the spatial econometric models, spatial proximity is more commonly used to represent the spatial attributes of each region [36], that is, two regions with spatial proximity are considered to have some correlation with marine fishery vulnerability in this study. If regions $i$ and $j$ have adjacent boundaries, the assigned value is 1, and 0 otherwise. The calculation formula is as follows:

$$w_{ij} = \begin{cases} 0 & , \quad \text{the two regions are not geographically adjacent} \\ 1 & , \quad \text{the two regions are geographically adjacent} \end{cases} \tag{8}$$

Geographical Distance Weighting Matrix ($W_d$)

The simple proximity weight matrix ignores the differences in the magnitude of the interactions between different regions due to different geographical distances; therefore, the role of geographical proximity in influencing the spatial correlation of the vulnerability of marine fisheries should be considered. To this end, a geographical distance weighting matrix is constructed using the inverse distance indicator [39], that is, the closer the geographical distance between two regions considered in this study, the more significant the spatial correlation of the vulnerability of marine fisheries and vice versa.

$$w_{ij} = \begin{cases} \frac{1}{d_{ij}^{2}} & , i \neq j \\ 0 & , i = j \end{cases} \tag{9}$$

where $d_{ij}$ is the distance between the provincial capital/municipal government of *i* province/municipality and *j* province/municipality.

Economic Distance Weighting Matrix ($W_g$)

The economic distance weight matrix describes the spatial difference in economic "distance" between regions to represent the difference in the level of economic development between regions [40]. In this study, the smaller the difference in the economic development level of fisheries between two regions, the closer the economic development of fisheries between the two regions and the more significant the spatial correlation of vulnerability of marine fisheries, and vice versa.

$$w_{ij} = \begin{cases} \frac{1}{|GDP_i - GDP_j|} & ,i \neq j \\ 0 & ,i = j \end{cases} \tag{10}$$

where $GDP_i$ and $GDP_j$ are real fisheries' *GDP* levels in regions *i* and *j* from 2009 to 2018, respectively.

### 2.2.4. Global Moran's I

Moran's I is a common test for spatial autocorrelation and is calculated as follows:

$$I = \frac{\sum\limits_{i=1}^{n}\sum\limits_{j=1}^{n} w_{ij}(x_i - \bar{x})(x_j - \bar{x})}{S^2 \sum\limits_{i=1}^{n}\sum\limits_{j=1}^{n} w_{ij}} \tag{11}$$

where *n* is the number of study subjects, $w_{ij}$ is the spatial weight matrix, $x_i$ and $x_j$ are the degrees of vulnerability of marine fisheries in regions *i* and *j*, and $S^2$ is the covariance of $x_i$ and $x_j$. Moran's I is located at $(-1, 1)$. A value above 0 indicates the presence of a positive spatial correlation in the vulnerability of marine fisheries; the larger its value, the more pronounced the spatial correlation. A value below 0 indicates a negative spatial correlation of marine fishery vulnerability, and the smaller its value, the more pronounced the spatial variability. When Moran's I is 0, it indicates that the vulnerability of marine fisheries shows a random distribution in space, and there is no spatial autocorrelation.

### 2.2.5. The Spatial Durbin Model

In this study, a fixed-effects SDM is selected for empirical analysis of the spatial effects of vulnerability in marine fisheries. The specific model settings are as follows:

$$vul_{it} = \alpha + \rho W vul_{it} + \beta X_{it} + \theta W X_{it} + \mu_i + \delta_t + \varepsilon_{it} \tag{12}$$

$vul_{it}$ is the vulnerability level of marine fisheries. $X_{it}$ is the factor affecting the vulnerability of marine fisheries and includes four dimensions: economic efficiency, industrial structure, environmental regulation, and ecological pollution. *W* is the weight matrix. $\alpha$ is the constant term. $\rho$ and $\beta$ are the spatial regression coefficients, and $\theta$ is the regression coefficient. $\mu_i$ and $\delta_t$ are individual and time effects, respectively. $\varepsilon_{it}$ is the random error.

### 2.2.6. The Spatial Spillover Effects

When the coefficient on the spatial lag term in the SDM is significantly non-zero, relying solely on the spatial lag coefficient in the SDM model leads to misinterpreting the estimation results. In other words, the spatial lag coefficients in this study cannot accurately reflect the spatial effects of the vulnerability of marine fisheries in different regions. Our results show significant spatial correlations in the vulnerability of marine fisheries between different regions. Changes in an influencing factor may cause changes in the vulnerability of marine fisheries in a province and may also impact neighboring regions. The direct

and indirect effects reflect this relationship. The direct effect is represented by the average degree of influence of the influencing factor in a region on the vulnerability of marine fisheries in that region, while the indirect effect means the average degree of influence of the influencing factor in a region on the vulnerability of marine fisheries in other neighboring regions. Drawing on LeSage and Pace's approach [41], Equation (12) can be transformed into Equation (13):

$$(I - \rho W)vul_{it} = (\beta X_{it} + \theta W X_{it}) + U \tag{13}$$

$$vul_{it} = \sum_{r=1}^{k} S_r(W)X_r + V(W)U \tag{14}$$

$$S_r(W) = V(W)(I\beta_r + \theta_r W) \tag{15}$$

$$V(W) = (I - \rho W)^{-1} \tag{16}$$

where $I$ is the unit matrix, and $U$ includes the remaining terms, such as the intercept term. Expanding Equation (14) yields Equation (17):

$$\begin{bmatrix} vul_{1t} \\ vul_{2t} \\ \vdots \\ vul_{nt} \end{bmatrix} = \sum_{r=1}^{k} \begin{bmatrix} S_r(W)_{11} & S_r(W)_{12} & \cdots & S_r(W)_{1n} \\ S_r(W)_{21} & S_r(W)_{22} & \cdots & S_r(W)_{2n} \\ \vdots & & \ddots & \vdots \\ S_r(W)_{n1} & S_r(W)_{n2} & \cdots & S_r(W)_{nn} \end{bmatrix} \begin{bmatrix} X_{1r} \\ X_{2r} \\ \vdots \\ X_{nr} \end{bmatrix} + V(W)U \tag{17}$$

where $S_r(W)_{ii} = \frac{\partial vul_{it}}{\partial X_{ir}}$ is the partial differential of $vul_{it}$ to $X_{ir}$, which measures the average effect of the $r$-th influencing factor of region $i$ on the vulnerability of the marine fisheries in that region, that is, the direct effect obtained by taking the mean of the diagonal elements of the matrix of Equation (17). $S_r(W)_{ji} = \frac{\partial vul_{it}}{\partial X_{jr}}$ is the partial differential of $vul_{it}$ to $X_{jr}$, which measures the average effect of the $r$-th influencing factor in region $j$ on the vulnerability of marine fisheries in region $i$, that is, the indirect effect obtained by taking the mean of the non-diagonal elements of the matrix of Equation (17).

## 3. Results

### 3.1. Evaluation of the Vulnerability of Marine Fisheries

3.1.1. Conceptual Framework and Indicator System

Based on the DPSR model, this study develops a conceptual framework for the vulnerability evaluation index system of marine fisheries in Chinese coastal provinces, as shown in Figure 1.

In this study, the GDP per capita and Engel's coefficient represent the economic drivers, whereas population density is a demographic driver. The regions affected by storm surge disasters, industrial wastewater discharges, and equivalent pollutant discharges indicate the ecological pressure on marine fisheries caused by human activities. Furthermore, the proportion of fishing production to total production and regions of culturable sea indicate the pressure on marine fisheries caused by the human exploitation of fishery resources. The environmental quality of the marine fisheries is expressed in terms of the water quality in offshore waters and economic losses from disasters as a proportion of the total economic output of fisheries, while the sustainability of resources for marine fisheries is expressed in terms of the diversity index, the number of aquatic fry per capita, and species richness. The level of social management is expressed by fishery law enforcement agencies, while the level of social governance is expressed by the governance capacity of industrial wastewater treatment facilities and the proportion of marine protected regions to the total sea region. The level of scientific and technological development is expressed by the investment in marine science and technology and the number of aquatic technology extension institutions. Based on the 18 indicators under the above four dimensions, a system of indicators for evaluating the vulnerability of China's marine fisheries is established, as shown in Table 1.

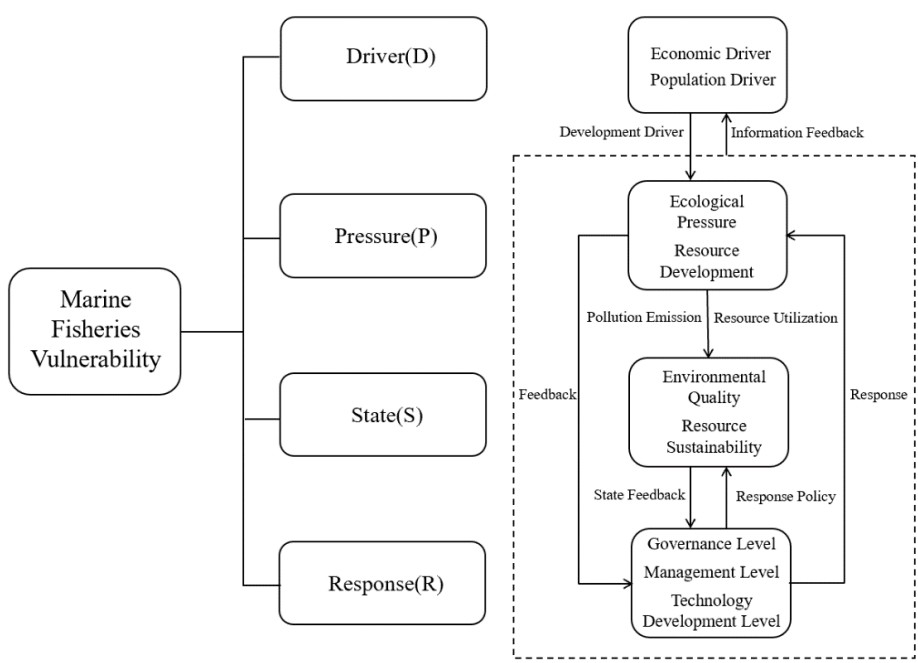

**Figure 1.** Conceptual framework for the vulnerability evaluation index system.

**Table 1.** Vulnerability evaluation indicator system for China's marine fisheries.

| Target Level | Sub-System | Metric Level | Properties |
|---|---|---|---|
| Driver (D) | Economic Driver | X1 GDP per capita | — |
| | | X2 Engel's coefficient | + |
| | Population Driver | X3 Population density | + |
| Pressure (P) | Ecological Pressure | X4 Storm surge disaster | + |
| | | X5 Industrial wastewater discharge | + |
| | | X6 Equivalent pollutant discharges | + |
| | Resource Development | X7 Fishing production/total production | + |
| | | X8 Region of culturable sea | — |
| State (S) | Environmental Quality | X9 Water quality in offshore waters | — |
| | | X10 Economic losses from disasters/total economic output of the fisheries | + |
| | Resource Sustainability | X11 Diversity index | — |
| | | X12 Number of aquatic fry per capita | — |
| | | X13 Species richness index | — |
| Response (R) | Governance Level | X14 Number of fisheries law enforcement agencies | — |
| | Management Level | X15 Governance capacity of industrial wastewater treatment facilities | — |
| | | X16 Region of marine protected regions/the total sea region | — |
| | Technology Development Level | X17 Investment in marine science and technology | — |
| | | X18 Number of aquatic technology extension institutions | — |

3.1.2. Evaluation Result

Further, based on the established marine fisheries vulnerability evaluation indicator system, this study applies the entropy method introduced above for a weighted average to calculate a comprehensive vulnerability score for the fisheries system of China's coastal provinces, and the scores are shown in Table 2.

**Table 2.** Extent of vulnerability of marine fisheries in the coastal regions of China.

| Year Province | 2009 | 2010 | 2011 | 2012 | 2013 | 2014 | 2015 | 2016 | 2017 | 2018 | Mean |
|---|---|---|---|---|---|---|---|---|---|---|---|
| Tianjin | 0.0056 | 0.0064 | 0.0061 | 0.0067 | 0.0072 | 0.0074 | 0.0073 | 0.0074 | 0.0076 | 0.0074 | 0.0069 |
| Hebei | 0.0057 | 0.0049 | 0.0051 | 0.0058 | 0.0045 | 0.0042 | 0.0044 | 0.0043 | 0.0044 | 0.0065 | 0.0050 |
| Liaoning | 0.0074 | 0.0095 | 0.0089 | 0.0106 | 0.0076 | 0.0069 | 0.0076 | 0.0065 | 0.0064 | 0.0084 | 0.0080 |
| Jiangsu | 0.0075 | 0.0057 | 0.0060 | 0.0110 | 0.0097 | 0.0084 | 0.0061 | 0.0075 | 0.0071 | 0.0072 | 0.0076 |
| Zhejiang | 0.0174 | 0.0146 | 0.0142 | 0.0161 | 0.0219 | 0.0167 | 0.0188 | 0.0170 | 0.0169 | 0.0175 | 0.0171 |
| Fujian | 0.0133 | 0.012 | 0.0111 | 0.0155 | 0.0149 | 0.0141 | 0.0168 | 0.0169 | 0.0123 | 0.0222 | 0.0149 |
| Shandong | 0.0070 | 0.0077 | 0.0097 | 0.0081 | 0.0119 | 0.0118 | 0.0100 | 0.0102 | 0.0102 | 0.0088 | 0.0095 |
| Guangdong | 0.0114 | 0.0102 | 0.0096 | 0.0110 | 0.0117 | 0.0111 | 0.0091 | 0.0087 | 0.0092 | 0.0121 | 0.0104 |
| Guangxi | 0.0064 | 0.0085 | 0.0065 | 0.0055 | 0.0103 | 0.0098 | 0.0044 | 0.0047 | 0.0045 | 0.0039 | 0.0065 |
| Hainan | 0.0110 | 0.0146 | 0.0082 | 0.0092 | 0.0091 | 0.0054 | 0.0061 | 0.0067 | 0.0078 | 0.0063 | 0.0084 |

## 4. Factors Influencing Vulnerability of Marine Fisheries

### 4.1. Global Spatial Autocorrelation Test

In this study, global Moran's I of the vulnerability of marine fisheries in 10 coastal regions in China from 2009 to 2018 was calculated using Stata software. The initial data on the vulnerability score of marine fisheries calculated above and the economic distance weight matrix were selected for testing. The results are presented in Table 3.

**Table 3.** Marine fisheries vulnerability global Moran's I.

| Year | 2009 | 2010 | 2011 | 2012 | 2013 | 2014 | 2015 | 2016 | 2017 | 2018 |
|---|---|---|---|---|---|---|---|---|---|---|
| Moran's I | 0.484 ** | 0.263 | 0.016 | 0.421 ** | 0.423 ** | 0.485 ** | 0.307 * | 0.419 ** | 0.215 *** | 0.451 *** |

Note: *, **, and *** indicate that the coefficient estimates are significantly non-zero at the 10%, 5%, and 1% levels, respectively.

### 4.2. Choice of Spatial Econometric Model

There are three standard spatial econometric models: the spatial autoregressive model (SAR), spatial error model (SEM), and the SDM. Therefore, the choice of the spatial econometric model needs to be made based on Moran's I test.

Using Stata for the test, from Table 4, LM(error), Robust LM(error), LM(lag), Robust LM(lag), *LR(SDM/SAR)*, *LR(SDM/SEM)*, and *Hausman* tests all passed significance verification.

**Table 4.** Choice of spatial econometric model.

| Variable | $W_{0,1}$ | $W_d$ | $W_g$ |
|---|---|---|---|
| | Test Values | | |
| *LM(error)* | 2.571 ** | 5.697 ** | 2.891 * |
| *Robust LM(error)* | 20.221 ** | 20.864 *** | 18.420 *** |
| *LM(lag)* | 10.497 *** | 24.778 *** | 17.504 *** |
| *Robust LM(lag)* | 10.148 *** | 39.945 *** | 33.033 *** |
| *LR—SDM—SAR* | 12.79 ** | 15.34 ** | 9.83 ** |
| *LR—SDM—SEM* | 12.54 ** | 15.22 ** | 8.72 ** |
| *Hausman* | 81.94 *** | 458.63 *** | 68.69 *** |

Note: *, **, and *** indicate that the coefficient estimates are significantly non-zero at the 10%, 5%, and 1% levels, respectively.

### 4.3. Influencing Factors Regression Results

Economic efficiency can influence the sustainability of marine fisheries through the input and output of fisheries resources [42], and the evolution of industrial structure can promote the flow and reallocation of fisheries resources and production factors in various industries [43]. Environmental regulation can reduce the mismatch of resources in the

industry and thus improve the productivity of the industry [44], and the construction of fisheries ecological civilization is the only way to sustainably develop fisheries [45]. Therefore, this study examines the spatial spillover effects of marine fishery vulnerability in four aspects of economic efficiency, industrial structure, environmental regulation, and ecological pollution. The scores calculated above represent the degree of marine fisheries vulnerability (*vul*), the size of economic efficiency (*gdp*) in terms of fisheries GDP, industrial upgrading rate (share of primary production in fisheries economy *1 + share of secondary production in fisheries economy *2 + share of tertiary production in fisheries economy *3) in terms of industrial structure upgrading (*ind*) [46]. The intensity of environmental regulation (*reg*) is expressed as the share of industrial wastewater treatment investment in the GDP [47]. The amount of direct wastewater discharged into the sea expresses the degree of ecological pollution (*pollu*).

This study uses Stata software for econometric analysis and regressions with individual, time, and double fixed effects. Empirically, the best results are obtained for time-fixed effects. The effect of each variable is the same under different spatial weight matrices; thus, our results are robust. The regression results are listed in Table 5.

**Table 5.** Regression results on factors influencing marine fisheries vulnerability.

|  | $W_{0,1}$ | $W_d$ | $W_g$ |
|---|---|---|---|
| *gdp* | 0.00014 * | 0.00044 ** | 0.00051 * |
|  | (0.63) | (2.21) | (2.87) |
| *ind* | −0.01591 *** | −0.01541 *** | −0.02494 *** |
|  | (−4.32) | (−3.68) | (−6.46) |
| *reg* | −0.00357 * | −0.00642 ** | −0.00031 |
|  | (−1.68) | (−2.02) | (−0.16) |
| *pollu* | 0.00013 ** | 0.00009 * | 0.00015 ** |
|  | (2.45) | (1.65) | (2.65) |
| $W \times gdp$ | 0.00049 ** | 0.00022 | 0.00020 * |
|  | (2.23) | (0.62) | (0.49) |
| $W \times ind$ | −0.00325 ** | −0.00188 | −0.02060 * |
|  | (−2.55) | (−0.23) | (−1.88) |
| $W \times reg$ | 0.00810 ** | 0.01495 *** | −0.00200 |
|  | (2.86) | (3.60) | (−0.27) |
| $W \times pollu$ | 0.00025 ** | 0.00060 ** | 0.00008 |
|  | (2.07) | (2.67) | (1.01) |
| $R^2$ | 0.1779 | 0.2543 | 0.0101 |

Note: *, **, and *** indicate that the coefficient estimates are significantly non-zero at the 10%, 5%, and 1% levels, respectively, and the values in parentheses are standard errors.

### 4.4. Spatial Spillover Effects of Vulnerability in Marine Fisheries

In this study, the spatial spillover effects of the vulnerability of marine fisheries in different provinces were estimated using Stata software, and the results are shown in Table 6.

**Table 6.** Influencing factors' spatial spillover effects.

|  | $W_{0,1}$ | | $W_d$ | | $W_g$ | |
|---|---|---|---|---|---|---|
|  | **Direct Effects** | **Spillover Effects** | **Direct Effects** | **Spillover Effects** | **Direct Effects** | **Spillover Effects** |
| *gdp* | 0.0002 ** | 0.0005 ** | 0.0004 ** | 0.0002 | 0.0005 *** | 0.0001 ** |
| *ind* | −0.0163 *** | −0.0035 | −0.0157 *** | 0.0001 | −0.0248 *** | −0.0182 * |
| *reg* | −0.0032 | 0.0072 *** | −0.0066 ** | 0.0146 *** | −0.0001 | −0.0018 * |
| *pollu* | 0.0001 ** | 0.0002 ** | 0.0001 | 0.0005 *** | 0.0001 ** | 0.0001 |

Note: *, **, and *** indicate that the coefficient estimates are significantly non-zero at the 10%, 5%, and 1% levels, respectively.

## 5. Discussion

### 5.1. Discussion of Evaluation Results

Considering the spatial dimension, the overall vulnerability of China's marine fisheries is characterized by a high level in the south and a low level in the north, which is consistent with the test results of the article by Li et al. [1]. As significant fishing provinces, Zhejiang, Fujian, and Guangdong have the most apparent vulnerability problems with their marine fisheries. Their long-term dependence on fishing to obtain the required resources has led to increasing vulnerability. Shandong, Hainan, Liaoning, Jiangsu, and Tianjin have serious vulnerability problems regarding their marine fisheries. Owing to their more developed fishery economies, more pollution is emitted during fishery production, putting more severe ecological pressure on marine fisheries. Guangxi and Hebei have less developed fishery economies because of their lack of well-developed marine fisheries and are not given as much importance in the national economy as other provinces. As a result, people do not have a strong interest in the exploitation and destruction of the fishery resource, and vulnerability estimates are optimistic. Among them, the results of the marine fisheries vulnerability assessment in the Bohai Rim region are consistent with those in the article by Gao et al. [48], further confirming the robustness of the paper's conclusions.

Under the time dimension, the 10 coastal regions can be divided into two dimensions based on the development trend of the vulnerability of marine fisheries in each region. First, Tianjin, Zhejiang, Fujian, and Shandong show fluctuating upward trends. Fujian has the most oversized upward trend, indicating that the vulnerability of marine fisheries here has become increasingly severe over time. The overall vulnerability of the fisheries in the Fujian Province is the second highest in China; therefore, the vulnerability of marine fisheries has become the most serious and urgent problem for the development of fisheries in Fujian. Hebei, Liaoning, Jiangsu, Guangdong, Guangxi, and Hainan show fluctuating downward trends. Hainan has the largest downward trend, falling from the fourth position in 2009 to the ninth in 2018, thus significantly improving vulnerability. This is followed by Guangxi, where the degree of vulnerability continues to decline, while vulnerability is viewed optimistically.

### 5.2. Discussion of Spatial Effects Results

5.2.1. Moran's I

From Table 3, the global Moran's I passed the significance test for eight years from 2009 to 2018, and all results showed positive spatial autocorrelation. In other words, the vulnerability of marine fisheries in the Chinese coastal regions is not randomly distributed in space and has the characteristics of an agglomerative distribution.

5.2.2. Discussion of Spatial Econometric Model

From Table 4, LM(error), Robust LM(error), LM(lag), and Robust LM(lag) are all significant at the 1% or 5% levels under the three different spatial weight matrices. Therefore, the spatial econometric model is more appropriate for testing than the ordinary least squares model. The LR (SDM/SAR) and LR (SDM/SEM) tests are significant at the 5% level, and the SDM could not be degraded to SAR/SEM. That is, SDM was more suitable than SAR and SEM for testing the spatial effects of vulnerability in marine fisheries. Finally, based on the result that the Hausman test is significant at the 1% level, using a fixed-effects model is superior to using a random-effects model. In summary, the SDM with fixed effects was selected for the empirical analysis of the spatial effects of vulnerability in marine fisheries.

5.2.3. Discussion of Regression Results

Table 5 shows that the regression coefficients on fisheries' GDP (*gdp*) are significantly positive at the 10% or 5% levels under the three spatial weight matrices. In other words, there is a significant positive relationship between fisheries' GDP and their vulnerability. An increase in their GDP can improve the wealth of fisheries. In turn, an increase in wealth levels can increase people's satisfaction with their material needs, which is a driving force for

productive human activities. Under the current production model, most economic growth in the fishing industry is quantitative rather than qualitative. In other words, economic growth is based on the exploitation of resources and the disposal of surplus resources, that is, the continuous extraction of resources from marine fisheries, their processing into the desired fishery products for consumption and use, and the disposal of surplus resources or products in the fishery ecosystem. As the cycle continues, the increasing GDP of the fishery industry is accompanied by the constant exploitation of resources and discharge of waste. Consequently, the vulnerability of marine fisheries increases with economic efficiency.

Table 5 shows that the regression coefficient on the industrial upgrading index (*ind*) is significantly negative at the 1% level for all three spatial weight matrices. In other words, there is a significant adverse effect between the degree of industrial structure rationality and the vulnerability of marine fisheries. On the one hand, the uncoordinated structure of the fishery industry leads to more primary products and less advanced processed products, which determines the blind exploitation of marine fishery resources to satisfy demand. If one seeks to satisfy one's interests to the detriment of protecting resources and ecology, there exist serious consequences that are difficult to remedy. Consequently, the vulnerability of marine fisheries will increase with the irrational structure of the industry. On the other hand, with the development of the social economy, the pursuit of leisure has gradually increased. The fishery economy is gradually shifting toward the tertiary industry, represented by the recreational fishing industry. Recreational fishing integrates traditional fisheries with ecological farming, sea picking, and leisure fishing. It is an emerging marine economic industry, in which fishery production and leisure are compatible under the same ecological and environmental conditions, which can enhance the economic benefits while considering the construction of ecological civilization. Therefore, the vulnerability of marine fisheries will continue to decrease with the increase in the rationalization of the industrial structure.

Table 5 shows that the regression coefficient on environmental regulation (*reg*) is significantly negative under the proximity weight matrix ($W_{0,1}$) and spatial-geographic weight matrix ($W_d$). In other words, there is a significant adverse effect between the intensity of environmental regulations and the vulnerability of marine fisheries. As the intensity of government environmental regulations increases, the original production methods no longer conform to current policies, and fishery enterprises are forced by government policies to transform and upgrade, which gradually reduces the negative environmental externalities brought by production. At the same time, companies spend a lot of money on environmental pollution testing and wastewater treatment equipment to meet environmental policy standards, and production funds come mostly from companies. To achieve production sustainability, the "learning by doing" effect allows companies to generate economies of scale in environmental management so that the benefits of production outweigh the costs of environmental management inputs and production technology and resource allocation efficiency increase. Consequently, the vulnerability of marine fisheries decreases as the intensity of environmental regulation increases.

Table 5 shows that the regression coefficients on ecological pollution (*pollu*) are significantly positive at the 5% and 10% levels for all three spatial weight matrices. In other words, there is a significant positive relationship between the degree of ecological pollution and the vulnerability of marine fisheries. The increase in human activity has led to many pollution sources flowing into the sea through different means. This includes elements such as nitrogen, phosphorus, and the industrial effluents discharged during production [49], which lead to severe heavy metal pollution and eutrophication of water bodies and the deterioration of water quality in offshore waters. On the one hand, the decline in water quality leads to a deterioration in the living environment of fisheries resources, which is detrimental to mariculture and aquatic fish species and thus poses a challenge to biodiversity. The decline in species is an essential factor that affects the sustainability of fisheries. On the other hand, water quality can also affect the productivity of marine fisheries. The productivity of marine regions with poorer water quality is lower for the same input cost,

and resources are overexploited to achieve economic benefits. The vulnerability of marine fisheries thus increases with the level of ecological pollution.

### 5.2.4. Discussion of Spatial Spillover Effects

Table 6 shows that an increase in regional fisheries' GDP has a positive direct effect on the vulnerability of a region's marine fisheries and a positive indirect effect on the vulnerability of those in surrounding regions. Taking the spatial proximity weight matrix as an example, an increase in fisheries' GDP would worsen the vulnerability of the region's marine fisheries by 0.0002 units, while worsening the vulnerability of those in surrounding regions by 0.0005 units. This may be because the economic benefits have led fishers in a region to stop being satisfied with the exploitation and use of their waters and to seek cross-regional cooperation to diversify their goods and services, thereby achieving higher economic benefits. The process of cross-regional cooperation can pose challenges to fishery resources and the systemic environment of the surrounding regions, thus increasing the vulnerability of marine fisheries.

Table 6 shows that upgrading the regional industrial structure has a negative direct effect on the vulnerability of a region's marine fisheries and on the vulnerability of the surrounding regions' marine fisheries. Taking the spatial economic distance weight matrix as an example, upgrading the industrial structure reduced the vulnerability of a region's marine fisheries by 0.0248 units. Comparatively, it reduced the vulnerability of those in surrounding regions by 0.0182 units. As previously mentioned, higher economic efficiency drives people to cooperate across regions. Moreover, upgrading industry structures can improve pollution emissions and the efficiency of the use of fishery resources through improved production methods and cooperation, thereby improving the degree of vulnerability of marine fisheries.

Table 6 shows that the intensity of regional environmental regulations has a negative direct effect on the vulnerability of marine fisheries in a region and a positive indirect effect on the vulnerability of those in surrounding regions. Using the spatial geographical weight matrix, for example, environmental regulation reduces the vulnerability of a region's marine fisheries by 0.0066 units, while increasing the vulnerability of those in surrounding regions by 0.0146 units. This may be because, when companies are subjected to a sudden environmental regulation policy, they do not have time to change their production methods to control pollution emissions. Therefore, to meet environmental standards, pollutants or polluting industries must be moved to the surrounding regions, which puts enormous environmental pressure on these regions, thereby increasing the vulnerability of marine fisheries.

Table 6 shows that an increase in regional pollution levels has a positive direct effect on the vulnerability of marine fisheries in a region and on the vulnerability of those in surrounding regions. Taking the spatial proximity weight matrix as an example, an increase in pollution will make the vulnerability of the region's marine fisheries 0.0001 units more severe, while making the vulnerability of those in surrounding regions 0.0002 units more severe. Pollutants from the production process flowing into the sea owing to the mobility of seawater will cause the contaminated sea region to shift geographically, which will harm the habitat of fishery resources in the surrounding regions, thus increasing the vulnerability of marine fisheries.

### 6. Conclusions

This study investigates the extent and influencing factors of the vulnerability of marine fisheries in the coastal regions of China and the spatial spillover effects of each influencing factor. To this end, it uses annual data from 2009 to 2018 for 10 coastal regions in China to construct a system of indicators for evaluating the vulnerability of marine fisheries in China's coastal regions by selecting 18 indicators under the four dimensions of the DPSR model: driver, pressure, state, and response. The degree of vulnerability of coastal marine fisheries in China is assessed using the entropy value method. From the evaluation results,

the overall vulnerability of marine fisheries in China's coastal provinces and municipalities is found to be characterized by a high level in the south and a low level in the north. Given the mobility of fishery resources and seawater, there is an intertemporal exchange of fishery resources and pollutants between provinces. Based on the evaluation of the vulnerability of marine fisheries in each region, this study further explores the factors influencing the vulnerability of marine fisheries in the coastal regions of China and their spatial spillover effects. This study establishes a spatial weight matrix based on spatial proximity, geographical distance, and economic disparity to ensure the robustness of the empirical results. The global autocorrelation test with the economic distance weight matrix show that the vulnerability of marine fisheries in coastal provinces and cities in China has positive spatial autocorrelation. Based on the LM, LR, and Hausman tests, the SDM with fixed effects is selected to study the influencing factors and spatial spillover effects on the vulnerability of marine fisheries in four aspects of economic efficiency, industrial structure, environmental regulation, and ecological pollution. The results show that: (1) there is a significant positive relationship between economic efficiency and ecological pollution and the vulnerability of marine fisheries. However, there is a significant negative relationship between industrial structure and environmental regulation and the vulnerability of marine fisheries. That is, the higher the economic efficiency and the more serious the ecological pollution, the higher the vulnerability of marine fisheries. The more reasonable the industrial structure and the stronger the level of environmental regulation, the lower the degree of vulnerability of marine fisheries. (2) Improved economic efficiency can increase the vulnerability of marine fisheries to the surrounding regions. Upgrading the industrial structure can reduce the vulnerability of marine fisheries in a specific region and the surrounding ones. An increase in the level of environmental regulations can reduce the vulnerability of marine fisheries in this region. However, this can increase the vulnerability of marine fisheries to the surrounding regions. An increase in the intensity of ecological pollution can increase the vulnerability of marine fisheries in a region and surrounding regions.

There are some limitations to this study. Owing to data availability, the data used are annual data at the district level. If quarterly or monthly data become available in the future, the results would be more accurate. More detailed data will thus be used to study the vulnerability of marine fisheries in the future. This research provides a theoretical basis and new perspective for the sustainable development of marine fisheries in China.

**Author Contributions:** Y.L. proposed the concept and the main steps of the research, including data sorting, experiment completion, and paper writing. J.J. conducted verification, review writing, and supervision. All authors have read and agreed to the published version of the manuscript.

**Funding:** This research was funded the National Natural Science Foundation of China (No.71873127), the National Social Science Foundation of China (No.19VHQ007), and Major Project of Social Science Planning of Shandong Province (No. 20AWTJ19).

**Institutional Review Board Statement:** Not applicable.

**Informed Consent Statement:** Not applicable.

**Data Availability Statement:** Not applicable.

**Conflicts of Interest:** The authors declare no conflict of interest.

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
