# Peer review of "Evaluation of Marine Fisheries Vulnerability in China and Its Spatial Effects: Evidence from Coastal Regions"

_agriculture, doi:10.3390/agriculture12060809_

Round 1

Reviewer 1 Report

The manuscript is poorly organized and does not follow journal instructions, the Material & Methods and Results are all blended together which makes the paper very hard to follow and understand. Analyses that were never mentioned initially in the “Methods” (such as the  spatial econometric methods) are then described as results. Please clearly separate the methods -description of data and analysis - from the results.

The results are incomplete and require presentation of the intermediate data analysis before you calculated the composite score for vulnerability-even as a Supplement. It is very hard to judge the analysis without looking at at least a descriptive view of the data and intermediate analysis.

Specific comments

Line 31-33, Implies that extensive fishery development is the cause of frequent natural disasters. Revise

Line 36-40, since there is such a rich and extensive literature in vulnerability of fisheries systems and coastal communities it is unecvcesarry to be this general, it is better to focus on more specific literature.

Lines 41-71. This background section should include relevant studies in the study region, such as the following. Please make sure all relevant studies are cited. 

Ding, Q., Shan, X., & Jin, X. (2020). Ecological footprint and vulnerability of marine capture fisheries in China. Acta Oceanologica Sinica, 39(4), 100-109.

Chen, Q., Su, H., Yu, X., & Hu, Q. (2020). Livelihood vulnerability of marine fishermen to multi-stresses under the vessel buyback and fishermen transfer programs in China: The case of Zhoushan City, Zhejiang Province. International journal of environmental research and public health, 17(3), 765.

Chen, Xinjun, and Qi Ding. "Global Climate Change and Sustainable Development of Fisheries." In Fisheries Resources Economics, pp. 323-346. Springer, Singapore, 2021.

Lines 193-198, citations or links for the data sources should be included.

Lines 206-220, this is background and introduction rather than methods. For a commonly-used method such as DPSR/DPSIR it is not necessary to describe the origin of the method, especially if it has already been previously published. Please revise to leave only the necessary information to understand the present study.

Table 5. Indicate what are the abbreviations in the Table caption or include the full name in column 1

Author Response

Dear reviewer:

Thank you for the opportunity to revise the manuscript (entitled “Evaluation of marine fisheries vulnerability in China and its spatial effects: Evidence from coastal regions”, agriculture-1732930). According to your advice, we amended the relevant part in the manuscript, and the revised text is highlighted in red in the manuscript. Our point-by-point answers to those comments and suggestions are set below.

We hope that the revised manuscript is now acceptable for publication. If you have any questions about this study, please do not hesitate to contact me.

Looking forward to hearing from you soon.

Sincerely yours,

Li Yutong, Ji Jianyue

Reviewer 2 Report

This paper sets out to assess (1) the degree of marine fisheries vulnerability in each region of China; (2) the factors causing such vulnerability, focusing on four broad factors: economic efficiency; ecological pollution; industrial structure; and environmental regulation; and (3) the spatial spill-over effects of marine fisheries vulnerability. The paper is well-written and clearly structured. It contains much mathematical analysis which I do not have the expertise to evaluate. I do, however, have four questions for the authors: 

First, why did they restrict their choice of factors affecting vulnerability to the above four factors? What about climate change? What about social divisions? What about power relationships? What about corruption?

 Second, their characterisation of ‘vulnerability’ appears confusing. For example, when they say (on lines 475-476; 479-480) that “there is a significant positive relationship between economic efficiency and ecological pollution and the vulnerability of marine fisheries”, evidently they mean that “the higher is the economic efficiency and the more serious the ecological pollution, the higher is the vulnerability of marine fisheries”. In other words, more economic efficiency and more ecological pollution means more vulnerability. So a ‘positive relationship’ means worse. Likewise, when the authors say (on lines 477-479; 481-482) that “there is a significant negative relationship between industrial structure and environmental regulation and the vulnerability of marine fisheries”, evidently they mean “The more reasonable is the industrial structure and the stronger the level of environmental regulation, the lower is the degree of vulnerability of marine fisheries”. In other words, more industrial structure and more environmental regulation means less vulnerability. So a ‘negative relationship’ means better.  This seems confusing.   

 Third, the above correlations seem too simplified. For example, higher economic efficiency may well increase vulnerability for some fishers (e.g., putting increased pressure on less efficient fishers) but it may also reduce vulnerability for other fishers (e.g., providing increased profit for more efficient fishers). Likewise, more environmental regulation might reduce vulnerability in some ways (e.g., protecting fish stocks from over-fishing) but it might increase vulnerability in other ways (e.g., by preventing fishers from switching from one target species to another as marine conditions change).     

 Fourth, on lines 182-184, the authors say “this study examines the spatial spill-over effects of marine fishery vulnerability in four aspects of economic efficiency, industrial structure, environmental regulation, and ecological pollution”. However, in section 4.4. Spatial Spill-over Effects of Vulnerability in Marine Fisheries, it is not explained whether increased spill-over means greater vulnerability or lesser vulnerability.   

 If the authors satisfactorily respond to these four queries, I would recommend publication of their paper. 

Author Response

(The authors gave the same response as above.)

Reviewer 3 Report

General comments

This paper tries to establish an evaluation index system for marine fisheries using the theoretical framework of the DPSR model. The entropy method was used to calculate the degree of marine fisheries vulnerability in 10 coastal regions in China from 2009 to 2018. The Spatial Durbin Model (SDM) was also used to analyse the influencing factors and spatial spillover effects of marine fisheries vulnerability from four perspectives of economic efficiency, industrial structure, environmental regulation, and ecological pollution. The objective of the manuscript, in general terms, is very interesting in the field of agriculture / fisheries studies, especially for an important country in fisheries such as China.

Although the paper has good objectives and shows some interesting results, it does not have a good structure (i.e., to provide more details about data used in methods section, and to clearly separate the results section) and lacks a discussion. The manuscript should pending suitable minor revision in light of the comments appended below.

More specific comments:

Title, abstract, keywords and format

-        Title, Abstract and Keywords: reflect the content of the paper.

-        Language: in general terms, the manuscript is well-written.

-        Structure / Suggested organisation of the paper:

1.      Subsection "2.5. Data Sources" (Lines 192–198), it is not clear what is the data collected and used for the study, especially for the readers who are not familiar with Chinese statistical yearbooks. Please provide more details about data used. For example, the authors stated in (Lines 490–491) that "There are some limitations to this study. Owing to data availability, the data used are annual data at the district level. If quarterly or monthly data become available in the future, the results would be more accurate". This is difficult to understand for the reader without providing details about the data used.

2.      I would change the order of this section and place it at the beginning of the methods section. It is very important before providing details about the analysis and methodologies used, to inform the readers about the data collected and for what reason it was collected. So, I would suggest organising the "2. Materials and Methods" section into 2 subsections: first "Data source and collection" and then "Data analysis".

3.      Honestly, starting from section "3. Evaluation of the Vulnerability of Marine Fisheries" the manuscript suffers mixing between methods and results. Please separate all results under a Results section.

4.      Please add a discussion section, before conclusions.

Figures:

-        Figure 1’ captions are very brief and don’t provide a good description of the figure.

Tables:

-        All tables’ captions are very brief and don’t provide a good description of the tables.

Conclusions

-        The section "5. Conclusions" provides a kind of a short summary of the previous sections (e.g., objective, methods used, results). However, the manuscript lacks a real discussion where the authors should use findings of other similar studies (e.g., papers with the same objectives; either using the same methodology or any other methodology) for comparison even for other regions and not necessarily only in China. For example, comparison with the results obtained by other methods such as PSA, and/or SICA. Also, based on the results, whether the methods used in this study (e.g, DPSR model) provide the promised results to close the gaps in the literature stream highlighted by the authors in line 73. What is then missing is a scientific/scholarly 'discussion' which links the proposed ideas in this paper into the other existing similar research.

Author Response

(The authors gave the same response as above.)
